# Enhanced Low-Energy Impact Localization for Carbon-Fiber Honeycomb Sandwich Panels Using LightGBM

**DOI:** 10.3390/s25247570

**Published:** 2025-12-12

**Authors:** Zifan He, Jiyun Lu, Shengming Cui, Chunhua Zhou, Yinuo Shao, Qi Wu, Hongfu Zuo

**Affiliations:** 1Civil Aviation Key Laboratory of Aircraft Health Monitoring and Intelligent Maintenance, Nanjing University of Aeronautics and Astronautics, Nanjing 211106, China; hezifan@nuaa.edu.cn (Z.H.); lujiyun@nuaa.edu.cn (J.L.); 2State Key Laboratory of Mechanics and Control for Aerospace Structures, Nanjing University of Aeronautics and Astronautics, Nanjing 210016, China; cuishengming@nuaa.edu.cn (S.C.); wu.qi@nuaa.edu.cn (Q.W.); 3Shanghai Institute of Satellite Engineering, Shanghai 201109, China; zhouchunhua@nuaa.edu.cn; 4College of Civil Aviation, Nanjing University of Aeronautics and Astronautics, Nanjing 211106, China; shaoyinuo3377@163.com

**Keywords:** honeycomb sandwich composites, low-energy impact localization, fiber Bragg grating sensor, layout optimization, machine learning

## Abstract

**Highlights:**

**What are the main findings?**
Optimized FBG-based monitoring method for accurate localization of low-energy impacts in honeycomb sandwich composites.Achieved high localization accuracy and real-time performance using feature- and data-parallel processing.

**What are the implications of the main findings?**
Scalable method supports real-time aerospace structure monitoring and maintenance.Findings provide guidance for enhancing structural load-bearing performance and preventing safety hazards.

**Abstract:**

Low-energy impacts have been demonstrated to cause damage and failure in aircraft structures, thereby affecting the structural load-bearing performance and creating safety hazards. In this study, an innovative damage-monitoring method based on a fiber Bragg grating (FBG) is proposed for honeycomb sandwich composites. The proposed method is applicable to honeycomb sandwich composites and integrates a light gradient boosting machine (LightGBM)-optimized impact localization method with feature-parallel and data-parallel processing in the machine learning architecture. An impact localization algorithm is applied to honeycomb sandwich composites using an array of multiplexed FBG sensors. The proposed algorithm exhibited substantial localization accuracy. The LightGBM method was employed to identify the optimal branching points for impact localization in real time, addressing the low-accuracy challenge in localizing low-energy impacts on the board structure when the fiber grating sensing system operates at a high sampling frequency.

## 1. Introduction

Carbon-fiber honeycomb sandwich composites combine lightweight design with high load-bearing capacity, vibration damping, and superior energy absorption, making them indispensable in modern aerospace structures. They are often used in aerospace applications, such as aircraft fuselage and wing structures. Under the influence of impact, such structures are prone to damage, such as core collapse and surface tearing, affecting structural performance and reliability. Low-energy impacts often cause internal damage that cannot be visually detected, posing significant challenges for structural health monitoring of these composites [1,2,3]. Damage resulting from low-energy impacts is not discernible through visualization methods, because damage manifestations such as matrix cracks and delamination primarily occur within the laminate. Most composites are brittle and absorb energy primarily through elastic deformation and damage mechanisms rather than plastic deformation. In a composite system, most impacts on the composite plate occur in the normal direction, and its resistance to impact damage is minimal because of the absence of a thickness-reinforced phase. Impact loads induce interlayer shear and tensile stresses that can lead to interlayer cracking. To address this challenge, a sensor network must be installed within the structure to measure the response signal upon impact. Through signal analysis, health monitoring technologies that can effectively localize impact damage and improve real-time damage detection in advanced composite structures can be developed.

Effective signal processing is fundamental to accurate impact localization, as it governs how structural-response data are translated into reliable damage-detection results. Researchers have investigated various localization algorithms, including Lamb waves [4], frequency-response functions [5], triangulation techniques [6], and artificial-intelligence algorithms [7]. Impact-response signal processing can be categorized based on the diverse research objects and methodologies employed, as follows:

The first category is based on time-difference localization techniques. The time-difference localization technique determines impact location based on the relationships among distance, angle, and wave propagation time. This is often accomplished using advanced signal-processing methods that extract the time-domain information of impact signals from various sensor locations. Ciampa and Meo [8] studied field impact detection, which can identify the location of acoustic emissions detected by the monitoring system in real time, and proposed an algorithm based on the surface combined with piezoelectric sensors to measure the difference in the stress wave in time–frequency analysis. Based on time–frequency analysis, the amplitude of the continuous wavelet transform was used to determine the arrival time of the wave packet, solving a set of nonlinear equations using a local Newton iterative method based on an unconstrained optimization technique to derive the shock position coordinates and wave velocity. Ren et al. [9] extended the spatial filter algorithm for impact monitoring in composite structures by utilizing two-dimensional linear piezoelectric sensor arrays for signal acquisition. They proposed a spatial filtering-based structural impact monitoring system that does not require wave-velocity estimation. This study proposes a spatial filter-based method for localizing structural impacts without the wave velocity.

The second category includes frequency-domain-based localization methods. Hiche et al. [10] proposed a novel localization method that involves measuring the maximum strain spectrum through fiber Bragg grating (FBG) sensors. FBG sensors are utilized to collect the strain signals and identify the maximum strain value corresponding to the impact location. This method requires only a limited structural analysis and a small number of sensors, and its feasibility was substantiated through simulation and experimental verification. The approach was validated through the simulation and experimental results.

The third category includes system modeling approaches. This approach entails a comparison between the measured and simulated signal features, where damage localization is determined by the minimum discrepancy between the measured and simulated structural signal features. Rezayat et al. [11] obtained data from FBG sensors and developed a variable-selective least-squares algorithm that utilizes structural patterns and data. This algorithm was experimentally validated and demonstrated three times the accuracy of the classical pseudo-inverse algorithm. Hafizi, Epaarachchi, and Lau [12] proposed a localization method for obtaining experimental data from FBG sensors using infrared sensing techniques. The group velocity was determined from the dispersion curves, and the difference between the crest times of the two sensors was defined as the time-delay difference. Then, localization was determined using a system of linear equations.

The fourth category includes machine learning-based and fitting techniques. This method requires establishing data input–output relationships for complex structures to achieve impact localization. Jang and Kim [13] proposed a low-energy impact localization method for a reinforced composite plate using reference data, trained the model using extensive reference data, and experimentally validated the proposed method. Shrestha et al. [14] proposed a low-energy impact localization method for a reinforced composite plate using a one-dimensional array of FBG sensors and a reference database. The proposed method was validated on a Jabiru UL-D wing, and the maximum error was found to be less than 35 mm, confirming the feasibility of this method. In a subsequent study, Shrestha et al. [15] proposed a method for localizing low-energy impacts based on anomalous error values. In this study, strain signals were acquired using FBG sensors, and validation was conducted on a carbon fiber-reinforced plastic (CFRP) prototype. The localization error was 10.7 mm. Sai et al. [16] proposed an FBG impact localization system based on quasi-Newton and particle swarm optimization algorithms. The FBG sensing network consists of eight fiber gratings, which are used for impact signal detection, with the time difference extracted using the Shannon wavelet transform. The impact localization system relied on nonlinear equations derived from the time difference and coordinates of the FBGs. These algorithms were employed to solve a set of nonlinear equations, yielding the coordinates of the impact source.

A complex nonlinear relationship exists between the impact response signal and low-velocity impact location, making it challenging to develop a mathematical model using traditional methods. Machine learning is a data-driven approach that constructs statistical models using training data, thereby characterizing nonlinear, high-dimensional, and high-complexity relationships among the data. Consequently, it has been extensively employed to identify low-velocity impact areas and localize low-velocity impacts on plate structures. Datta et al. [17] extracted impact features, including peaks, means, standard deviations, and energy indices, from impact response signals. These features were extracted using a least-squares support vector regression (SVR) model, which was used to identify the locations of low-velocity impacts on CFRP structures. To locate the low-energy effect on the laminate, Lu et al. [18] extracted the band energy corresponding to the sixth node as the impact feature and optimized the hyperparameters of the SVR model. This was achieved through wavelet packet decomposition of the impact response signal. Sai et al. [19] developed an impact localization model based on an extreme learning machine utilizing the band energies corresponding to the first and second nodes. Furthermore, Lu et al. [20] used the wavelet method to eliminate noise in an impact response signal, thereby enhancing the precision of extracting the time difference of arrival (TDOA). They then incorporated TDOA as an input feature into a least-squares support vector machine to detect the location of low-velocity impacts on CFRP laminates. Yu et al. [21] extracted the short-time energy feature and combined it with an optimized support vector machine (SVM) model to achieve accurate low-velocity impact localization for CFRP laminates. Zheng [22] introduced a method for impact localization on honeycomb sandwich panels using a projective dictionary pair-learning classifier, showcasing advancements in impact detection technology. A comprehensive review of the extant literature indicates increased research efforts on the impact resistance of honeycomb sandwich composites and the optimization of their design for low-energy impact scenarios. Further research is required to explore diverse materials and structural configurations to enhance the impact resistance of these panels. Xu et al. [23] explored the interaction between corrosion and mechanical loads on steel plates using discrete and distributed fiber-optic sensors. Their study revealed how FBG sensors could effectively monitor corrosion-induced damage, while optical frequency-domain reflectometry (OFDR) sensors provided high spatial resolution for distributed sensing. Recent advancements in FBG-based systems have focused on their integration with various sensing technologies to enhance localization accuracy and reliability in complex environments. Significant research has been conducted on integrating FBG sensors with other sensing modalities, such as piezoelectric sensors and accelerometers, to improve performance in dynamic applications. Ping et al. [24] explored the integration of FBG sensors with smart materials, which has shown promising results in enhancing the sensitivity of impact detection systems. This work is particularly relevant, as it addresses the challenges of multi-modal sensor integration, which is a crucial aspect of our method.

Although numerous machine-learning approaches have been developed for impact localization, most still suffer from limited accuracy, overfitting on small datasets, and inadequate feature representations for low-energy impacts in complex honeycomb structures. These drawbacks include the difficulty in obtaining stable input datasets, susceptibility to underfitting and overfitting, the need to collect a large number of training samples for the regression model, and the selection of appropriate hyperparameters to enhance its regression capability. Moreover, current impact localization methods based on machine learning primarily focus on extracting individual features, such as time-, frequency-, and time–frequency-domain features, as inputs to the regression model. This approach disregards the significance of multi-domain features in enhancing the efficacy of impact localization methods. To overcome these limitations, this study introduces a LightGBM-based impact localization framework specifically tailored to the anisotropic mechanical behavior and signal characteristics of carbon-fiber honeycomb structures. The proposed algorithm integrates gradient-based unilateral sampling with mutually exclusive feature binding. It utilizes time-domain signals with reduced sample data for impact localization, thereby reducing the required number of samples. Additionally, it employs FBG sensors for impact localization with reduced interference from electromagnetic signals. This improves model efficiency while reducing the number of feature dimensions, thereby preventing overfitting. The proposed method enables fast and accurate impact localization in honeycomb sandwich structures.

Building upon this motivation, the present study integrates FBG sensing with LightGBM learning to enhance low-energy-impact localization accuracy for honeycomb composites through combined experimental and computational validation. It methodically integrates signal-processing technology, feature-extraction methods, and machine-learning methods to develop a low-energy impact localization method for composite honeycomb boards. It addresses the challenge of low accuracy in the low-energy impact localization in board structures when utilizing a high-frequency sampling fiber grating sensing system. First, a low-energy impact experimental system for honeycomb sandwich composites was constructed, and a method based on empirical modal decomposition was explored to eliminate trend components from the impact response signal. The time-domain, frequency-domain, and time–frequency-domain features were extracted and integrated. The wavelet-packet characteristics of the impact response signal were analyzed using the wavelet-packet decomposition method. A novel impact feature extraction method was developed, and the impact region was identified based on a set of wavelet packet features. The impact features and regions were established. The identification method models the relationship between impact features and the distance from the low-energy impact to the FBG sensor. Additionally, a low-energy impact localization method for honeycomb sandwich composites based on a LightGBM-based model is proposed. Furthermore, the performance of the impact localization method was evaluated, as illustrated in Figure 1. The remainder of this paper is organized as follows: Section 2 presents the experimental setup, Section 3 details the proposed algorithm, Section 4 discusses the results and validation, and Section 5 presents the conclusions.

## 2. Experiments

This study used a carbon-fiber aluminum honeycomb sandwich panel (Shanghai Institute of Satellite Engineering, Shanghai, China) with assembly holes measuring 300 mm × 300 mm × 15 mm. The upper and lower surfaces of the sandwich panel consisted of 1-mm-thick T700/AG80 carbon fiber cladding, and the core layer was a hexagonal aluminum honeycomb with a wall thickness of 0.3 mm. A schematic of the test piece is shown in Figure 2. The carbon fiber used in this study was T700, with an elastic modulus of 135 GPa, a density of 1.58 g/cm^3^ and a Poisson’s ratio of 0.27. The honeycomb core was made of aluminum, with a density of 0.025 g/cm^3^, a Poisson’s ratio of 0.33 and a thickness of 15 mm. The core’s cell size was approximately 15 mm, and the core material’s elastic modulus was 3.39 × 10^−5^ GPa. The tested area, excluding the edge cladding and assembly holes, was 255 mm × 255 mm. The measurement area was divided into square grids with a side length of 15 mm, resulting in 256 grid intersections. Impacts were applied at grid intersections. According to the working conditions of the actual composite honeycomb panel, the panel was secured using its assembly holes to simulate the loading of structural components during the operation of an actual aircraft.

The FBGs (SHENHUA OPTOCAL, Zibo, China) utilized in this study were composed of SMF-28 fibers with polyimide coatings, operating within the wavelength range of 1510–1590 nm, at temperatures ranging from –40 to 120 °C, with a grating length of 10 mm. The impact localization performance depends significantly on the spatial arrangement of the FBG sensors. In this work, the six-sensor layout shown in Figure 3 was designed through a knowledge- and test-driven process rather than a formal mathematical optimization algorithm. The following aspects were considered: (1) the geometric symmetry of the carbon-fiber honeycomb sandwich panel; (2) the expected propagation directions of the dominant guided waves in the face sheet; and (3) the preliminary impact under different candidate layouts.

Six FBG sensors were affixed to the back of the specimen with the geometric center as the origin and the horizontal and vertical centerlines as the X and Y axes, respectively. The six sensors were affixed with the numbering shown in Table 1; the specific arrangement is shown in Figure 3. FBG5 and FBG6 were both positioned at (0, 0) to capture symmetric reference strains at the geometric center, enabling calibration and correction of signal drift near the assembly hole.

Impact localization is performed by analyzing the response signals received by the FBG sensors placed on the composite panel. Honeycomb sandwich composites exhibit anisotropic wave propagation characteristics, where the wave speed and signal attenuation vary depending on the material orientation. Moreover, assembly holes can significantly alter the local wave propagation characteristics. In particular, stress concentrations near the holes can cause wave scattering, reflection, and diffraction, which may complicate accurate impact localization. To mitigate this issue, our method incorporates a geometric-aware sensor layout that strategically places FBG sensors in positions that maximize sensitivity to these local disturbances. Additionally, we developed a correction mechanism that adjusts for the local wave propagation anomalies near assembly holes by incorporating a set of features that account for the proximity to these regions. This approach ensures that the impact localization remains robust, even in the presence of complex geometric features such as holes.

Figure 4 illustrates a localization device comprising an impact signal-generating system, including an impact pendulum and hammerhead, used to apply impacts to the test specimen, with adjustable pendulum length and hammerhead weight and size. The test specimen measured 300 mm × 300 mm × 15 mm, the pendulum length was l=400 mm, the hammerhead diameter was 15 mm, and the combined pendulum weight was  m=55 g.

In this study, the term “low-energy impact” is used in a practical sense to describe impact events that do not cause visible surface damage on the carbon-fiber honeycomb sandwich panels but can still generate internal stress waves and barely visible impact damage. Quantitatively, all impacts are produced by a drop-weight (or pendulum) device with impact energy:(1)E=mgh,
where m represents the impactor mass and h represents the vertical drop height. In this experiment, $h=l\times \left(1+sin60{}^{\circ}\right)=600$ mm.

In this study, we ensured precise impact positioning by adjusting the impact hammerhead’s position vertically and horizontally. This was achieved using the impact position adjustment device, which comprised a control rail, displacement restrictor, and component for adjusting the optical axis. The impact energy adjustment device angle limiter controls the impact energy by varying the swing angle. For the release heights used in our experiments, the corresponding impact energies are E=mgh=0.33 J. This is within the range typically associated with low-energy impacts on composite structures, in which internal damage may occur, while the external surface remains almost unchanged. Therefore, throughout this paper, “low-energy impact” specifically refers to impacts within this experimentally defined energy range.

The test site plan is shown in Figure 5, where the impact position adjustment device, including the height position adjustment slide, lateral position adjustment optical axis, and height limit device, was used to adjust the impact position of the impact hammerhead up and down, and left and right, to ensure the accuracy of the impact position. The impact energy adjustment device, specifically the pendulum height angle limit device, used the pendulum height angle to control the impact energy. The FBG sensor array under consideration in this study was connected to Beijing Tongwei Technology Co., Ltd.’s (Beijing, China) Gator demodulation system, operating within the wavelength range of 1515–1585 nm and temperature range of –20 to 55 °C. This system was employed to collect changes in the center wavelength of the FBG sensors at a sampling frequency of 19.23 kHz.

## 3. Localization Algorithm

### 3.1. Experimental Setup and Signal Acquisition

When using fiber-optic grating sensors for the impact localization of the structure to be tested, a plurality of fiber grating sensors is uniformly arranged on the structure to be tested, forming a sensor network. Recently, deep learning-based approaches, such as convolutional neural networks (CNNs), have been explored for intelligent damage detection [25]; however, they often require extensive datasets and incur high computational cost. To address this, gradient boosting techniques such as LightGBM offer lightweight and efficient alternatives, particularly in small-sample scenarios. Furthermore, time–frequency-domain features extracted via wavelet packet decomposition [26] and optimized sensor layouts [27] have been proven effective in enhancing localization performance, which motivates the sensor design and feature engineering strategies used in this study. The accurate localization of low-energy impacts in composite structures is a critical challenge in structural health monitoring. Despite the proposal of various localization algorithms in the literature, including those based on Lamb waves and piezoelectric sensors, these methods frequently encounter difficulties in detecting low-energy impacts. This is attributed to the weak signals generated by such impacts, which often mask the signals required for precise detection. The necessity for a refined methodology that can accurately localize low-energy impacts in complex composite materials is the impetus behind the development of the approach presented in this study.

Before acquiring FBG sensor data for impact localization, it is necessary to establish a dataset. During the acquisition of each impact dataset, the impact test grid resolution was 1 mm, and three identical impacts were performed at 256 grid intersections to record the center-wavelength changes in the six FBG sensors.

A substantial body of research has demonstrated that optimal sensor placement enhances positioning accuracy and robustness. Research by Sai et al. [16] further indicates that increasing the number of sensors improves the robustness of low-energy impact positioning systems. Yue and Sharif Khodaei [28] examined the impact of different measurement strategies on recognition accuracy, and their results supported the necessity of multidirectional sensor deployment. As a result of the induced propagation wave, the center wavelength of the fiber grating is shifted, the strain signals corresponding to the impacts collected by a plurality of fiber grating sensors on the structure to be tested are obtained, and the time-domain signals of each fiber grating sensor are obtained for each FBG sensor for the 5 s of the pendulum fall. Figure 6 shows the time-domain impact response signals obtained from the six FBG sensors when the point near the honeycomb plate mounting hole (–75 mm, 75 mm) is hit, recording the time-domain signals from 10 ms before impact arrival to 50 ms after. The wavelength–strain transformation relationship is 1 µε = 1.07 pm at room temperature. The strains measured by FBG5 and FBG6 at the panel center were smaller (≤50 µε), consistent with the expected stress-wave attenuation pattern, validating the central calibration sensors. In comparison, FBG1, FBG2, FBG3, and FBG4 in the periphery exhibited larger strains, and the recorded signal waveforms and envelopes were clear.

The FBG sensors were utilized to measure the strain signals, and the proposed method exhibited high accuracy in impact localization. It is acknowledged that mechanical noise can have a substantial impact on the performance of FBG sensors, particularly in dynamic environments. The presence of mechanical vibrations may introduce interference, which can compromise the accuracy of the localization results. To mitigate the impact of mechanical noise on the FBG signals, signal-processing techniques such as filtering and denoising were applied. The implementation of these measures enhanced the system’s robustness and facilitated precise impact localization. Regarding temperature effects, given that the experiments in this study were conducted with high-speed demodulation, the temperature variation during this brief period was negligible. Consequently, temperature fluctuations did not exert a substantial influence on the strain signal during the experimental process, and temperature compensation was deemed unnecessary in this particular instance.

For each of these 256 grid intersections, three identical impacts were performed to capture the variability and account for the natural variability in impact behavior. Thus, a total of 768 impact points (256 × 3) were generated for the dataset. Each experimental dataset contained 768 samples. In addition to maximum and minimum strain values, supplementary time–frequency features such as wavelet packet energy and short-time energy were extracted to capture shock-wave propagation characteristics and enhance feature diversity. Figure 7 provides a visual representation of several signal wavelet packet energy decomposition features.

In this study, features for the impact localization task were extracted from multiple domains: they included time-domain features (e.g., peak values, mean, standard deviation), frequency-domain features (e.g., frequency components, spectral energy), and time–frequency-domain features (e.g., wavelet packet energies). These features were selected based on their relevance to the signal characteristics of low-energy impacts. Specifically, time-domain features were chosen for their ability to capture the transient nature of impact signals, frequency-domain features were selected to identify oscillatory behavior, and time–frequency-domain features were included to track dynamic changes in the impact signals over time. To ensure that only the most relevant features were used and to avoid redundancy, we performed feature importance analysis using a correlation-based feature selection approach. This method helped to identify and remove highly correlated or redundant features. Additionally, we applied recursive feature elimination (RFE) in combination with a LightGBM model to evaluate the importance of each feature. Features with low importance scores were eliminated to reduce the risk of overfitting and improve model generalization. As a result, the final set of features used in the model contained those that most significantly contributed to impact localization performance.

To eliminate directional blind spots and enhance impact localization accuracy, this study symmetrically arranged two additional sensors (FBG5 and FBG6) at the plate center, with their optical fibers oriented perpendicular to each other (at 90°). Given the sensitivity of FBG sensors to axial strain, single-direction placement may result in the creation of sensing blind spots when confronted with shock waves propagating from different directions. Orthogonal placement has been demonstrated to significantly enhance multidirectional strain capture capability, thereby increasing sensitivity and coverage for changes in impact direction. According to the extant literature, orthogonal configurations have been demonstrated to markedly curtail directional errors and enhance signal feature integrity in composite materials and anisotropic structures. The configuration of the sensors is illustrated in Figure 8.

Previous low-energy impact localization algorithms for composite plates have two significant limitations. First, they require a substantial amount of training data, which can be computationally intensive due to the optimization of the model parameters and the selection of multi-domain features. To address these challenges, this section presents a novel low-energy impact experimental system for composite plates. The system utilizes the wavelet packet decomposition method to analyze the relationship between the wavelet packet characteristics of the impact response signal and the location of low-energy impacts. Additionally, it defines a novel method for extracting impact features. Research has been conducted on impact-region identification methods based on the wavelet packet feature set and impact localization methods based on the stochastic fractal search algorithm. A method that integrates wavelet packet recognition and impact localization was proposed to identify impact regions and determine impact locations. The impact localization method involves a two-step localization process for low-energy impacts on composite plates based on wavelet packet features.

### 3.2. Algorithm Implementation—LightGBM

LightGBM is a gradient boosting framework proposed by Ke et al. [29] for computing classification, regression, and other problems. Because of its one-sided sampling based on the gradient and fusion binding of mutually exclusive features, LightGBM significantly reduces runtime and memory usage compared to models such as categorical boosting (CatBoost), X (Extreme) Gradient Boosting (XGBoost), and the gradient boosting decision tree (GBDT). LightGBM enhances computational efficiency through gradient-based one-sided sampling and feature exclusivity binding, which significantly reduce memory usage and training time while maintaining prediction accuracy. To reduce the number of features while keeping all the information, the LightGBM model adopts the principle of feature mutual exclusion binding, which implies that the high-dimensional data is represented sparsely and enables the fusion of mutually exclusive features. The decision-tree splitting strategy in this model is a histogram algorithm that differs from other classification and regression algorithms in that the grow-by-leaf strategy in LightGBM has a depth restriction, calculates the splitting gain of all leaves, and splits the leaf with the most significant increase, allowing for efficient operation while mitigating overfitting. Figure 9 shows the LightGBM model structure and experimental workflow.

In Figure 6, the strain values recorded by different FBG sensors exhibit significant variations, primarily due to their spatial locations relative to the impact point. The observed strain differences indicate the varying degrees of wave propagation across the structure, and these variations can be used to infer the impact location. The LightGBM model acquires the capability to correlate these variations with specific impact locations through its training process, enabling more accurate impact localization. The LightGBM model has been engineered to process intricate and nonlinear relationships, rendering it particularly well-suited to address the variability in strain data across sensors. In the training process, LightGBM acquires knowledge regarding the manner in which the strain values from diverse sensors, which are subject to variation based on their proximity to the impact point, contribute to the overall impact localization. The decision-tree structure of LightGBM enables the model to compartmentalize the feature space and thereby capture the interactions between strain values at differing sensor locations. The model’s capacity to simulate nonlinear relationships facilitates its effective management of strain variations and precise prediction of impact locations. Specifically, the LightGBM model employs strain differences as pivotal features to estimate the impact location. It is evident that by acquiring knowledge of the spatial patterns of strain variation across the sensors, the model can efficiently map the strain data to the impact location. The model’s high accuracy in localization is primarily attributed to its capacity for integrating strain data from multiple sensors at varying distances from the impact point.

Yue [28] compared the accuracy of least-squares support vector machines (LSSVMs) and artificial neural networks (ANNs) for impact detection in aeronautical applications, demonstrating that LSSVMs outperform ANNs when training data is limited. Compared to the traditional machine-learning composite honeycomb panel impact localization system, network intrusion impact localization based on deep learning exhibits a significant improvement in accuracy. Multilayer perceptrons (MLPs) [30], CNNs [31], recurrent neural networks (RNNs) [32], long short-term memory (LSTM) [33], and other related algorithms have been applied to intrusion detection systems. We selected LightGBM for impact localization because it has several key advantages that make it particularly well-suited for this task. LightGBM is a gradient boosting framework offering high computational efficiency, particularly with large datasets. It has been shown to outperform many traditional models in terms of speed and accuracy. Additionally, ability to handle large feature sets and reduce overfitting through built-in regularization makes it ideal for our complex, high-dimensional impact localization problem. We evaluated other potential models, including ensemble methods such as XGBoost and Random Forest, as well as deep-learning approaches such as CNNs and RNNs. However, after testing these models under the same experimental conditions, we found that LightGBM provided the best balance among model accuracy, training time, and generalization ability. For example, although deep-learning models, such as CNNs and RNNs, achieved promising results, they required significantly more training data and computational resources, rendering them impractical for real-time impact localization. In contrast, LightGBM demonstrated excellent performance with a relatively small training dataset and shorter inference time, making it well-suited for our real-time monitoring environment. Furthermore, LightGBM’s ability to handle both numerical and categorical features without extensive preprocessing facilitated integration with our experimental setup.

In this study, LightGBM was selected for impact localization owing to its efficiency in handling large datasets with a relatively small number of training samples. In contrast to traditional machine-learning models, LightGBM uses gradient boosting decision trees, which allows it to effectively manage high-dimensional data and interactions between features. The model’s regression formula was chosen because it can capture the nonlinear relationships between the impact response and the sensor readings, which are essential for accurately localizing the impact point. To tailor LightGBM for our specific problem, we made several adjustments to the model’s hyperparameters. For example, the learning rate was set to 0.05 to prevent overfitting while maintaining fast convergence. The number of leaf nodes was optimized to balance model complexity and prediction accuracy, ensuring that the model captured intricate relationships in the data without becoming too complex. We also experimented with the number of boosting rounds (set to 1000) and tree depth (set to 6) to improve the model’s ability to generalize and prevent underfitting.

Feature interactions were carefully selected to include time-domain, frequency-domain, and time–frequency-domain features, which are directly related to the physical characteristics of the sensor signals. By including these diverse feature types, the model leveraged the complementary information provided by each domain to achieve more accurate localization. In this section, a regression model is constructed using LightGBM implemented in Python 3.13.0, with the strain maxima and minima of the six FBG sensors as inputs to the model and the output target variable y as the coordinates where the impact point is located. To ensure statistical robustness, a 10-fold cross-validation scheme was adopted, and both the mean RMSE and 95% confidence intervals were examined to assess model generalization.

During the training of the model, the mean squared error (MSE) function was employed as the loss function to optimize the parameters. MSE is frequently employed in regression problems and effectively penalizes substantial errors, which is crucial for precise impact localization. In addition to the MSE, the root-mean-square error (RMSE) and mean absolute error (MAE) were used to assess the model performance. 

### 3.3. Impact Localization Model Updating

The impact test was designed to simulate real-world structural installation conditions within the assembly hole. As a result, stress-wave conduction becomes irregular, leading to higher strain values in the region adjacent to the assembly hole than in other areas. Strain gauge FBG sensors were incorporated into the fittings to mitigate positioning errors caused by the assembly hole. After recalibrating the central sensors (FBG5 and FBG6) and optimizing peripheral sensor layout, the localization accuracy was improved by 35–50%, confirming the effectiveness of the geometry-aware placement strategy (by comparison in Figure 10).

Figure 10 presents a comparison between models that have been trained with four and six FBG sensors, under various sensor layouts. Here, “model_initial” refers to the model trained using the initial four-sensor configuration, and “model_updated” refers to the optimized sensor layout model incorporating FBG5 and FBG6.

## 4. Validation of Localization Results 

Figure 11 shows the nonbaseline locations designated for validation purposes. The total number of validation points was 60, and the proposed algorithm was used for impact localization. The localization errors shown in Figure 11 were obtained from 60 physically measured validation points (test set), representing the actual impact locations used for spatial accuracy evaluation. Compared to the grid size of 15 mm, the proposed LightGBM model achieved a maximum localization error of 4.24 mm and an average error of 1.40 mm, demonstrating a substantial improvement in spatial accuracy over conventional regression models. Despite the variation in the number of pre-measured signals, the findings not only demonstrate improved localization accuracy but also confirm that the proposed LightGBM-based approach effectively balances model complexity and generalization, offering a practical framework for structural health monitoring. These results validate the proposed algorithm, highlighting its ability to achieve high localization precision and computational efficiency in low-energy impact scenarios.

To optimize the LightGBM model, we performed a grid search over several key hyperparameters. We applied 10-fold cross-validation to the training data, and for each combination of hyperparameters, the MSE was computed. The optimal hyperparameters were selected based on the best MSE performance during cross-validation. This approach helps prevent overfitting while ensuring that the model generalizes well to unseen data. The final optimized hyperparameters selected from this process are presented in Table 2.

This section analyzes the performance of the LightGBM model in localizing low-velocity impacts in composite plates. To this end, a comparative analysis was conducted between the LightGBM model and three regression models: CatBoost, XGBoost, and GBDT. To ensure a fair comparison, the optimal subset of features obtained from the LightGBM model served as the input for all regression models. The number of multi-domain features corresponding to the horizontal coordinates of the shock was 255, whereas the number of multi-domain features corresponding to the vertical coordinates of the shock was 256. Each test was conducted independently 15 times.

Table 3 presents the mean absolute error (MAE) and root-mean-square error (RMSE) of the predicted coordinates of the 15 random shocks obtained from each model. Figure 10 presents the localization error of each random shock obtained from the LightGBM model and the three contrasting regression models.

In contrast, the MAE and RMSE values presented in Table 3 were derived from multiple randomized validation trials, where four algorithms (LightGBM, XGBoost, CatBoost, and GBDT) were tested using datasets with intentionally increased variation to evaluate robustness under diverse signal conditions. Because of the larger variability in these trials, the numerical errors (e.g., MAE = 5.8 mm) were expectedly higher than the mean spatial error (1.40 mm) from the 60-point physical test set.

As shown in Table 3, the LightGBM model achieves the smallest RMSE (10.2 mm) and MAE (5.8 mm) among all the tested models, outperforming XGBoost, CatBoost, and GBDT with statistical significance (*p* < 0.05). This indicates that the observed performance differences are unlikely to have occurred by chance and are statistically meaningful. The experimental findings demonstrated that the mean localization error when the LightGBM algorithm was employed was 5.8 mm, addressing the challenge of impact signal localization in composite honeycomb sandwich panels.

As indicated by Table 3 and Figure 12, the LightGBM algorithm outperforms baseline boosting models by reducing both the MAE and RMSE, demonstrating stronger generalization capability due to its leafwise growth and histogram-based split strategy. The localization error, maximum localization error, minimum localization error, and average localization error for each of the 15 random shocks obtained from the LightGBM model were considerably smaller than those obtained from the CatBoost, XGBoost, and GBDT models.

In summary, when the same subset of optimal features is used as input to the regression model, the LightGBM model provides higher localization accuracy for random shocks in the monitoring area of honeycomb sandwich composites compared to the three other regression models. In Figure 13, the LightGBM predictions closely align with measured impact coordinates, confirming the model’s stability and robustness under varying impact locations.

Figure 13 shows the model’s positioning performance in actual testing, with the spatial distribution of errors presented through an error heatmap. The model demonstrates high positioning accuracy in the central region, while errors are more pronounced in the peripheral areas. These large-error predictions were primarily caused by the reduced sensitivity of the sensor network in those regions, as well as the potential for wave propagation anomalies, such as interference or attenuation, in areas with complex geometries (e.g., near assembly holes or edges). Additionally, the model’s performance can be affected by noise or signal distortion, which may be caused by mechanical vibrations or external environmental factors. To better understand these failure cases, we analyzed the error distribution and found that the majority of large errors occurred in the areas where the sensor response was relatively weak or less distinct. To address this, we are exploring methods to improve the sensor layout, increase the sensor coverage, and integrate additional features such as wave propagation characteristics into the model to account for these challenging conditions. The findings suggest that the model’s performance is subject to variation across different zones.

To evaluate the real-time applicability of the proposed LightGBM-based localization system, we measured the model’s average inference time. On a standard desktop CPU (Intel Corporation, Santa Clara, CA, USA) (Intel i7-11700, 2.5 GHz, 16 GB RAM), the average inference time per impact event was approximately 3.2 ms, excluding data acquisition and preprocessing time. Given that the FBG interrogator operates at a sampling frequency of 19.23 kHz (0.05 ms per sample), the model’s inference time does not exceed the typical signal-acquisition rate, supporting its potential use in real-time structural health monitoring applications. Regarding scalability, we tested the model’s ability to handle large-scale data by simulating a monitoring system with *N* sensors. We found that the model’s performance remained stable even as the number of sensors increased, with only a slight increase in processing time as sensors were added. To further enhance scalability, we implemented a parallelized version of the model, allowing for faster processing in distributed systems.

## 5. Conclusions

In this study, a low-energy impact testing platform for carbon-fiber aluminum honeycomb sandwich panels was established to accurately localize impact positions and enhance deformation monitoring through an integrated LightGBM–FBG approach. The impact time-domain strain signals were analyzed, and the results indicated that the maximum strain recorded by FBG sensors accurately reflected the impact position on honeycomb sandwich panels. The proposed LightGBM-based model demonstrated superior localization accuracy compared to that of conventional boosting methods while significantly reducing the training time and computational load, using optimized model parameters to determine impact positions.

Although this study focused on square sandwich panels, the method can potentially be extended to non-square or irregular structures through sensor reconfiguration and geometry-adaptive training, which will be validated in future work. By adjusting the sensor configuration and partitioning the monitoring area, the process can be adapted to localize irregular composite structures. Although a geometric modeling approach requires extensive training data, the recognition-based approach offers enhanced versatility. The proposed approach overcomes the limitations of traditional TDOA techniques by learning nonlinear signal-to-distance relationships via LightGBM, thus achieving higher localization precision without explicit wave-speed estimation and resolving the interference between the signal amplitude and mounting distance due to the assembly problem. Experimental validation confirmed that the method effectively localizes impacts on honeycomb composites under multi-point constraints, indicating its engineering applicability for aerospace structure monitoring and maintenance. In future work, the proposed framework can support predictive maintenance strategies by automatically triggering localized inspection only when impact events exceed critical thresholds, thus optimizing NDT efficiency and reducing operational costs.

Recent deep learning-based methods, such as CNNs and RNNs, have achieved remarkable performance in localizing low-energy impacts by automatically extracting relevant features from sensor data and predicting impact locations with high precision. Additionally, generative adversarial networks have been explored to augment training data and improve model robustness in low-data regimes. These methods excel at handling large amounts of data and complex, nonlinear relationships between sensor readings. Furthermore, physics-informed approaches, such as physics-informed neural networks, integrate physical laws into the learning process to enforce consistency with wave propagation models. This enhances model performance in realistic, noisy environments where traditional data-driven models may fail. Hybrid approaches, which combine deep learning with physics-based models, have also exhibited considerable promise, leveraging the advantages of both data-driven learning and physical constraints to achieve accurate and robust localization.

In conclusion, the proposed method demonstrates a robust framework for low-energy impact localization in honeycomb sandwich panels. However, several challenges remain in extending the method to more complex scenarios, such as higher-energy impacts and different material types, and meeting the computational demands associated with training the model on new structures. We believe that exploring transfer learning and few-shot learning, along with optimizing computational processes, will be key to overcoming these challenges and improving the model’s scalability and generalizability. Future work will focus on these aspects, with the aim of making the method more versatile and applicable to a wider range of structural health monitoring applications.

## Figures and Tables

**Figure 1 sensors-25-07570-f001:**
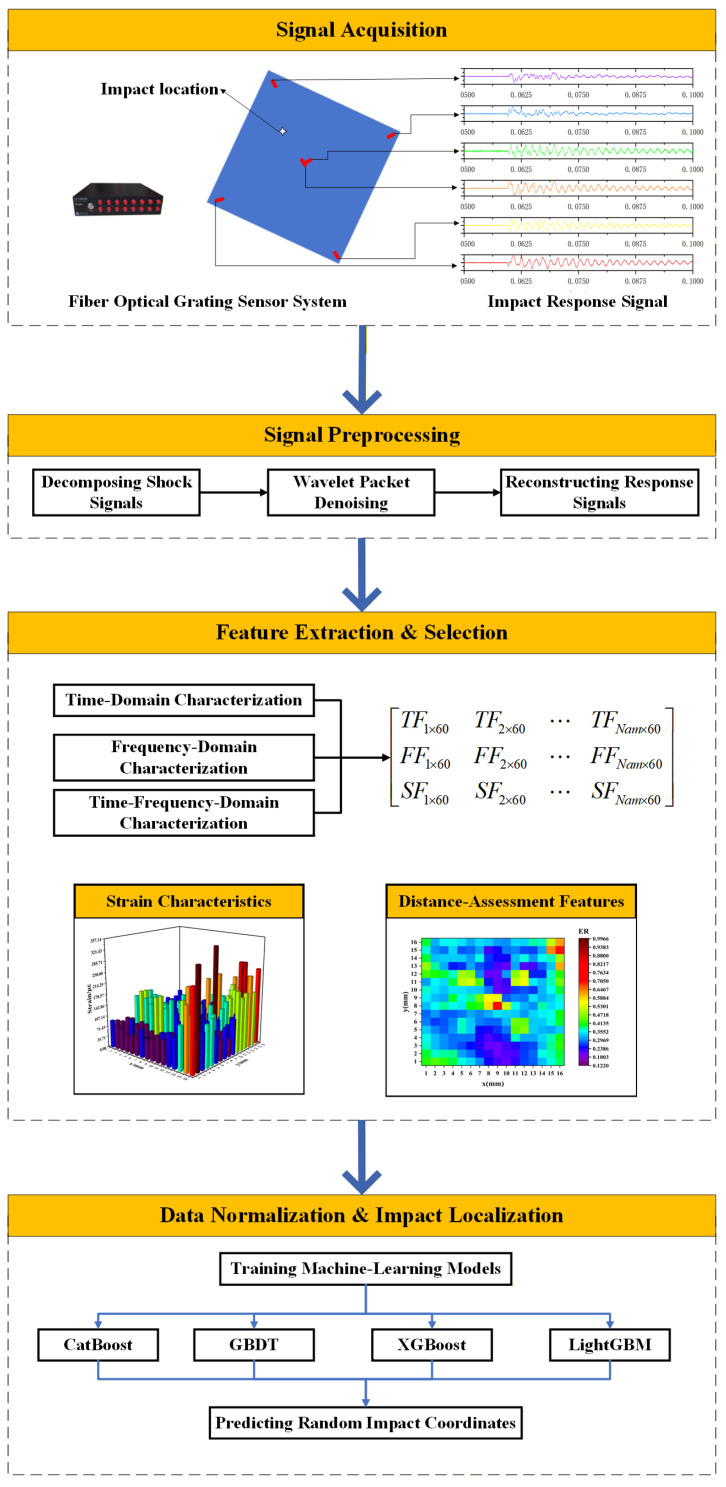
Flowchart of the low-energy impact localization method using the machine-learning model with multi-domain features for a honeycomb sandwich composite panel. *TF_n_*_×60_ refers to the time-domain features extracted from the shock signal, where “*n*” indicates the number of samples per window, and “60” refers to the number of time-domain features extracted from a 60-sample window. *FF_n_*_×20_ refers to frequency-domain features derived from the signal’s fast Fourier transform and *SF_n_*_×60_ refers to time–frequency-domain features calculated from the short-time Fourier transform.

**Figure 2 sensors-25-07570-f002:**
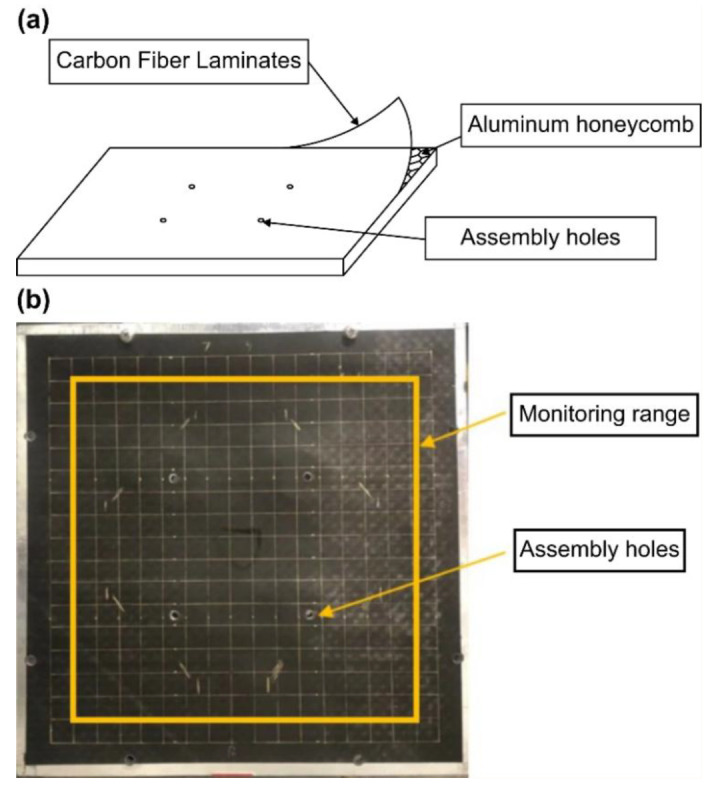
Schematic of the test piece. (**a**) Schematic of the test piece. (**b**) Specimen.

**Figure 3 sensors-25-07570-f003:**
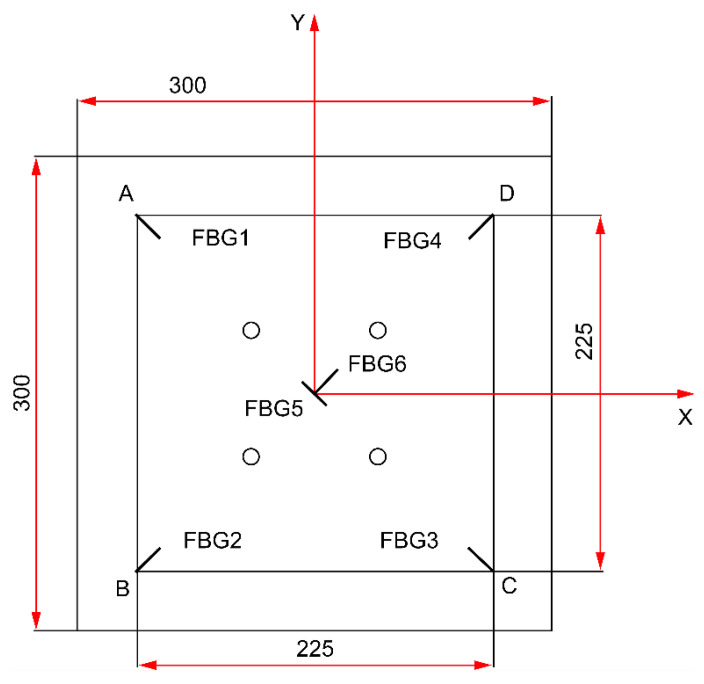
Sensor layout.

**Figure 4 sensors-25-07570-f004:**
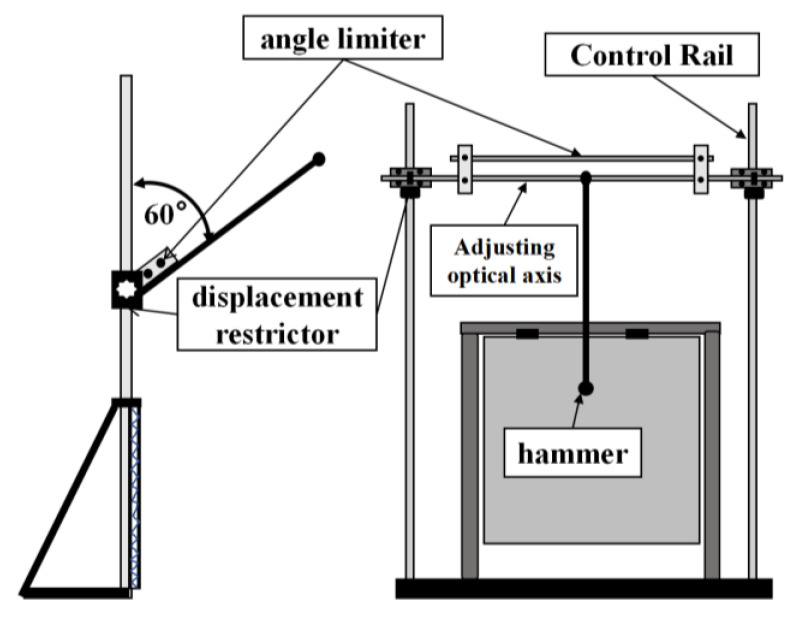
Schematic of the impact loading device.

**Figure 5 sensors-25-07570-f005:**
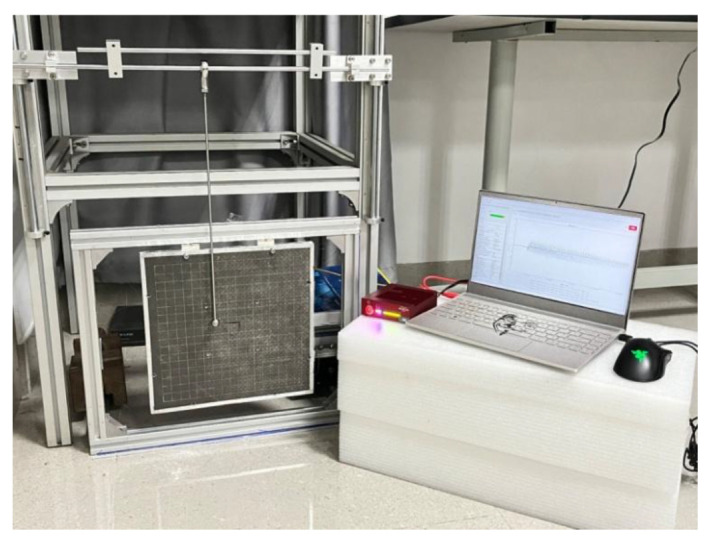
Experimental setup for the impact loading device.

**Figure 6 sensors-25-07570-f006:**
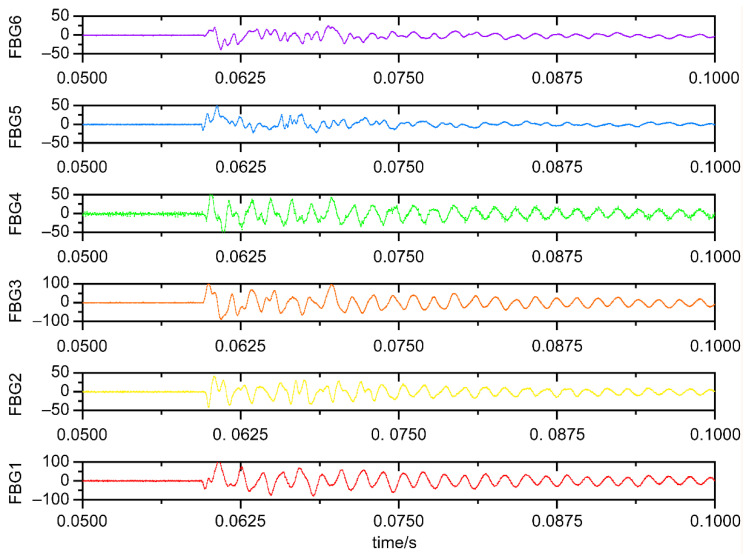
Measured impact signals from a commercial high-speed FBG interrogator.

**Figure 7 sensors-25-07570-f007:**
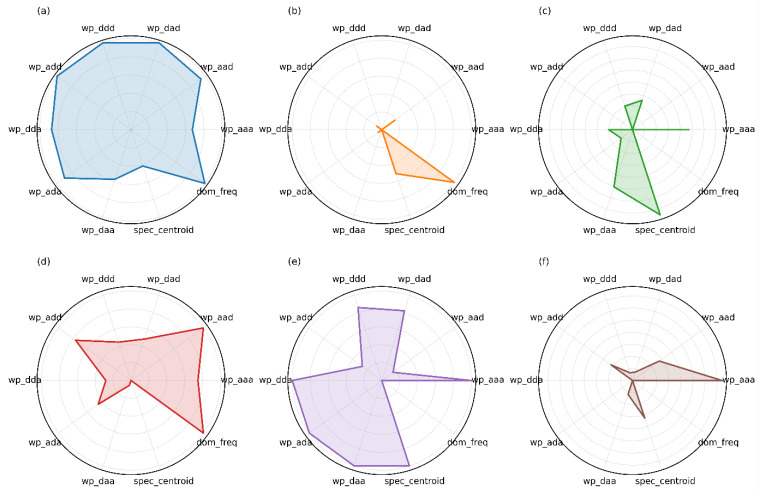
Wavelet packet energy decomposition features of the six FBG sensors. (**a**–**f**) Feature vectors of FBG1–FBG6, respec-tively. Note: “wp_xxx” corresponds to the energy of a signal in a specific frequency subband. The path name consists of a/d (a = low frequency, d = high frequency); for example, “wp_add” refers to energy in the low–high–high frequency subbands.

**Figure 8 sensors-25-07570-f008:**
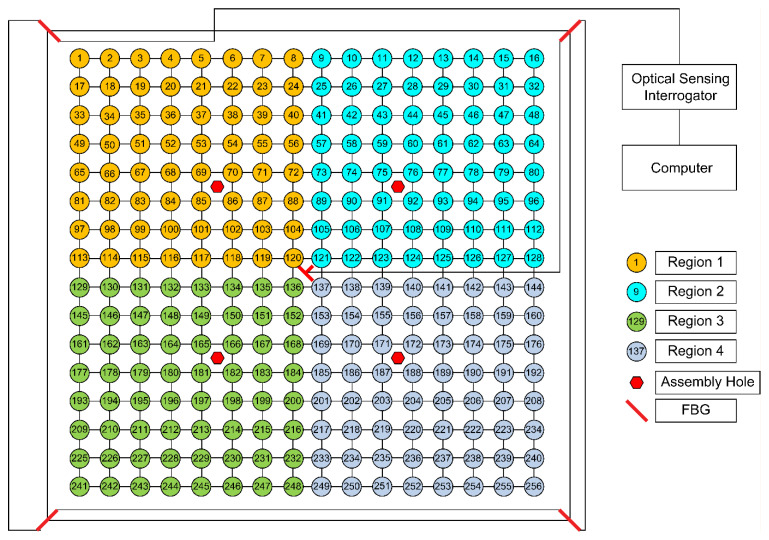
Schematic of composite honeycomb panel low-energy impact signal feature processing.

**Figure 9 sensors-25-07570-f009:**
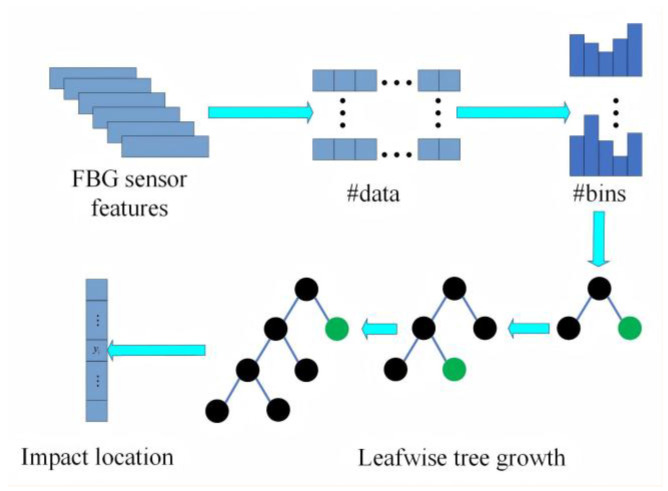
LightGBM model structure and experimental flowchart. In this figure, the symbol “#” indicates “number of”. Specifically, #data represents the number of data samples and #bins represents the number of bins.

**Figure 10 sensors-25-07570-f010:**
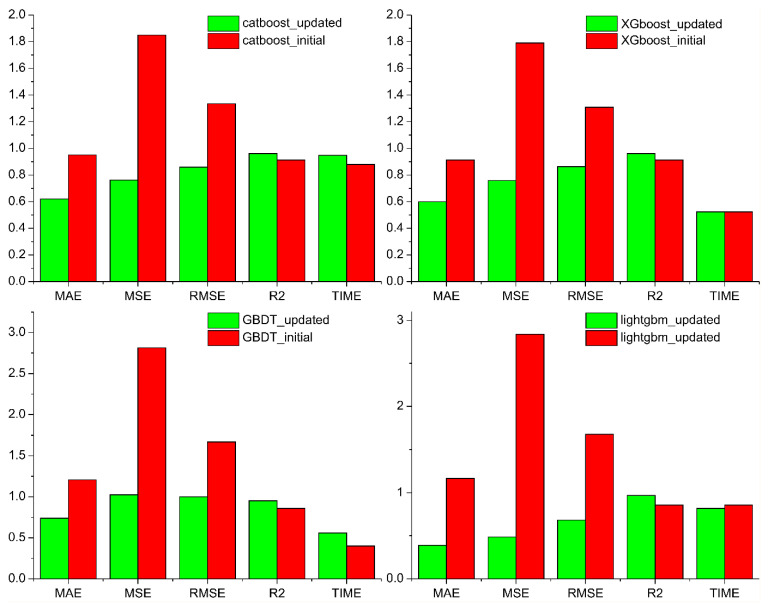
Comparison of the accuracy of machine-learning model sensor layout optimization.

**Figure 11 sensors-25-07570-f011:**
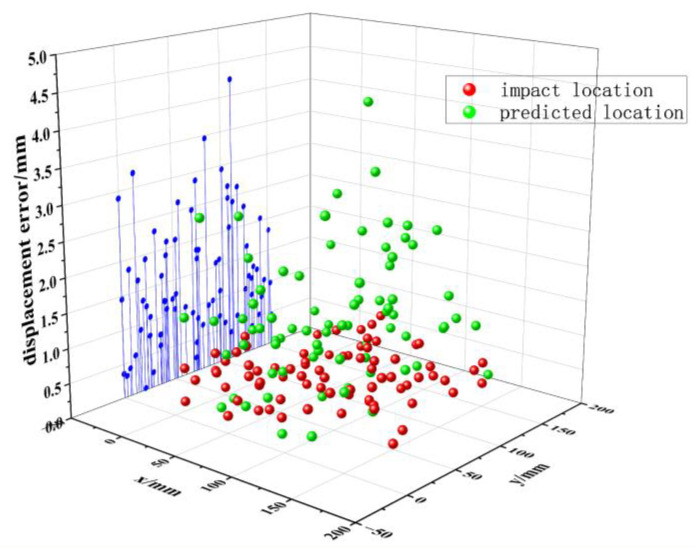
Comparison of the predicted and actual locations.

**Figure 12 sensors-25-07570-f012:**
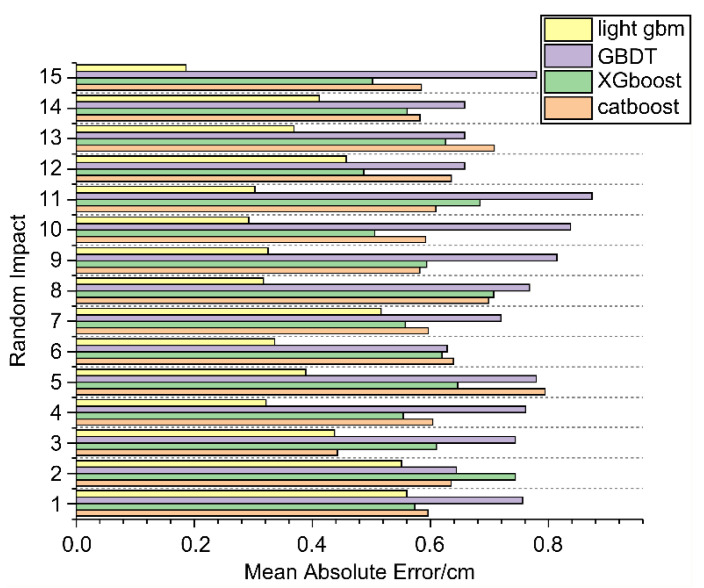
Localization-error analysis of the four regression models.

**Figure 13 sensors-25-07570-f013:**
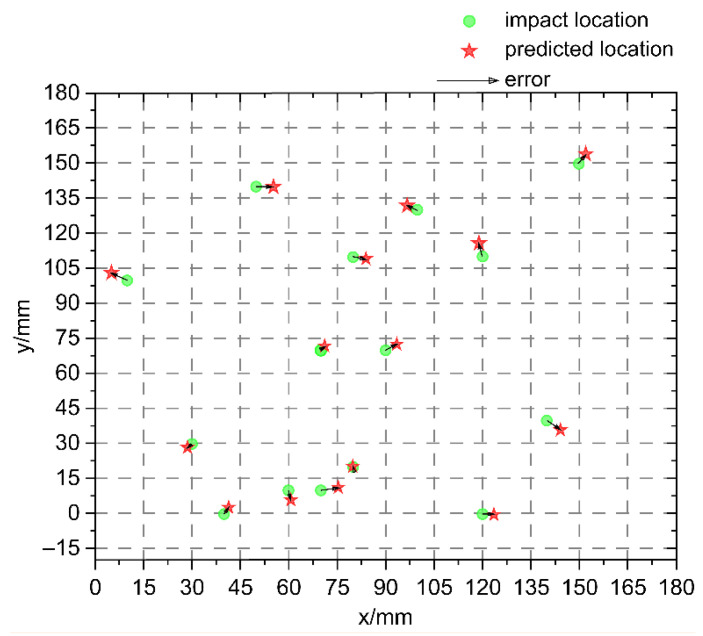
Low-velocity impact localization results of the honeycomb sandwich composites based on LightBGM modeling.

**Table 1 sensors-25-07570-t001:** FBG sensor center wavelength and position.

FBG	Center Wavelength λ/nm	Localization /mm
FBG1	1555	(−112.5, 112.5)
FBG2	1550	(−112.5, −112.5)
FBG3	1535	(112.5, −112.5)
FBG4	1560	(112.5, 112.5)
FBG5	1530	(0, 0)
FBG6	1545	(0, 0)

**Table 2 sensors-25-07570-t002:** LightGBM optimal parameters.

Parameter	Optimal Value
max_depth	5
num_leaves	31
learning_rate	0.05
reg_alpha	0.01
reg_lambda	0.1
min_child_weight	2
colsample_bytree	0.9
subsample	0.9

**Table 3 sensors-25-07570-t003:** Algorithm model accuracy comparison.

ML Model	MAE/mm	RMSE/mm
LightGBM	5.8	10.2
XGBoost	10.4	13.2
CatBoost	9.3	19.2
GBDT	11.1	11.4

## Data Availability

Data will be made available upon reasonable request.

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
