# Peer review of "Enhanced Low-Energy Impact Localization for Carbon-Fiber Honeycomb Sandwich Panels Using LightGBM"

_sensors, 2025, doi:10.3390/s25247570_

Round 1
Reviewer 1 Report
Comments and Suggestions for Authors
This manuscript presents an approach for low-energy impact localization in carbon-fiber honeycomb sandwich panels using Fiber Bragg Grating (FBG) sensors and the LightGBM algorithm. The study combines multi-domain feature extraction with a gradient-boosting framework to address challenges in localization accuracy and real-time performance. While the proposed method demonstrates promising results, several fundamental issues undermine its academic rigor and practical applicability. The experimental validation appears limited in scope, and the technical approach lacks depth in addressing the anisotropic and heterogeneous nature of honeycomb structures. The following points highlight specific concerns that should be addressed to improve the manuscript's contribution and credibility. Overall, it is an interesting work, and can be considered for publication after revision.
-Why was the specific arrangement of FBG sensors chosen, and how was the potential influence of sensor placement on directional sensitivity and blind zones systematically evaluated?
-The manuscript does not clarify how the multi-domain features were selected or whether feature importance analysis was performed to avoid redundancy and overfitting.
-How does the proposed method account for the anisotropic wave propagation characteristics inherent in honeycomb sandwich composites, especially near assembly holes?
-The study lacks a detailed discussion on the influence of impact energy variation on localization accuracy, which is critical for real-world applications.
-Why was LightGBM preferred over other ensemble or deep learning models, and were alternative architectures evaluated under the same experimental conditions?
-The experimental validation is limited to a single panel geometry and size, which restricts the generalizability of the findings to other structural configurations.
-The manuscript does not adequately address the effect of environmental factors, such as temperature or mechanical noise, on FBG signal stability and model performance.
-In addition, important FBG integrated researches should be reviewed as background, for example, DOI: 10.1016/j.sna.2024.115422
-There is insufficient analysis of the model's computational efficiency and scalability for large-scale or real-time structural health monitoring applications.
-The authors should provide more insight into the failure cases or large-error predictions and discuss potential reasons for such inaccuracies.
-The comparison with existing methods is superficial and does not include recent deep learning-based or physics-informed approaches for impact localization.
Please double check english presentations.
Reviewer 2 Report
Comments and Suggestions for Authors
Comments:
- This paper explores how to use fiber Bragg grating (FBG) sensors for impact location and monitoring. However, it lacks an introduction to the application of FBGs in impact monitoring in the Introduction. Authors may refer to the following literature: https://www.sciencedirect.com/science/article/abs/pii/S0030399224000112.
- In Section 2, material properties of the carbon fiber layers and the honeycomb core (such as elastic modulus, density, and core size) that have not been reported should be included to allow for comparison with other studies and to enable numerical reproduction.
- In Section 3.1, the description mixes experimental details, literature background, and methodological arguments, resulting in a lack of focus. Separating the initial research motivation from the specific experimental procedures would make this section clearer and more concise.
- In Section 3.2, A significant portion of the text repeats the general theory of LightGBM and comparisons with other machine learning models; this can be simplified. Instead, the emphasis should be placed on how LightGBM is tailored to this specific collision localization problem—for example, why the regression formula was chosen, how feature interactions relate to physical signal features, and how to select or adjust model hyperparameters (learning rate, number of leaf nodes, number of boosting rounds, tree depth, etc.).
- In Section 4, this section reports several error metrics (maximum positioning error, mean error, MAE, RMSE), but these values ​​are not always consistent and are not interpreted in relation to the dataset (e.g., the mean error reported earlier is 1.40 mm, while the MAE reported later is 5.8 mm). Please indicate whether these metrics were calculated under different test conditions, datasets, grid resolutions, or validation phases. Providing a unified table summarizing all error metrics calculated under consistent conditions will help avoid confusion.
Reviewer 3 Report
Comments and Suggestions for Authors
The paper presents an interesting approach to low-energy impact localization in carbon-fiber honeycomb sandwich panels using FBG sensors and a LightGBM model. The integration of signal processing, feature extraction, and machine learning for real-time, accurate impact detection is a valuable contribution to structural health monitoring. The paper, entitled "Enhanced Low-Energy Impact Localization for Carbon-Fiber Honeycomb Sandwich Panels Using LightGBM," is interesting and worth disseminating. However, several aspects of the methodology, results presentation, and discussion require significant refinement to meet the standards of a technical publication. A major revision is recommended, with particular attention to the following points:
- Clarity on "Low-Energy Impact" is needed; the paper frequently mentions "low-energy impacts" but does not provide a clear definition or a range of energy levels considered. This is crucial for understanding the scope and applicability of the proposed method. Please define what constitutes a "low-energy impact" in the context of this study (e.g., impact energy in Joules).
- Detailed experimental setup description is missing, such as:
- FBG sensors are mentioned but not discussed their parameters, specific details about the FBG interrogator (e.g., sampling frequency, wavelength range, resolution) are important for reproducibility and evaluating the sensor's suitability.
- Provide more information on how the impact energy was controlled and calibrated. "Pendulum height angle" is mentioned (line 229), but details on how this translates to specific impact energies are missing.
- Describe the data acquisition system used (hardware and software) in more detail.
- The paper mentions optimizing the sensor layout (lines 205-211, Figure 3, Figure 8). While the orthogonal placement of FBG5 and FBG6 is explained, a more comprehensive justification for the entire six-sensor layout, especially FBG1-FBG4, is needed. Was a formal optimization process used, or was it based on prior knowledge?
- Signal pre-processing details is missing; the flowchart (Figure 1) shows "Decomposing Shock Signals," "Wavelet Packet Denoising," and "Reconstructing Response Signals." More technical details on each step are required, including the specific wavelet used, decomposition levels, and denoising parameters.
- How feature extraction and selection has been made, pls elaborate, such as:
- Clearer feature definitions; such as: feature matrix (Figure 1) shows TFnx60, FFnx20, SFnx60. Please clearly define what these abbreviations stand for and what each number (e.g., 60, 20) represents (e.g., number of features, time windows).
- Motivation for feature choice; Justify the selection of time-domain, frequency-domain, and time-frequency-domain features. Why are these specific features chosen, and how do they contribute to impact localization?
- "Strain Characteristics" and "Distance Assessment Features"; elaborate on these two boxes in Figure 1. What specific characteristics and features are extracted here, and how are they used?
- Explain in depth the LightGBM Model details, such as:
- Hyperparameter tuning; as optimal parameters are listed in Table 2, describe the hyperparameter tuning process in more detail (e.g., search space, optimization algorithm).
- Loss function and evaluation metrics; clearly state the loss function used for LightGBM and why RMSE and MAE were chosen as evaluation metrics.
- Dataset description needs improvement; clarify the number of unique impact points used for training and validation. The text states "256 grid intersections" and "three identical impacts were performed at 256 grid intersections" (line 246), and "total number of validation points was 60" (line 363). Reconcile these numbers and explain the dataset split (training vs. validation). And data augmentation; If any data augmentation techniques were used to increase the dataset size, describe them.
- Discussion of signal waveforms (Fig. 6) is needed; such as: the text (lines 261-266) discusses the strain levels of different FBG sensors. Expand on the implications of these differences for impact localization and how the LightGBM model handles this variability.
- Refer to Figure 7; Provide a more detailed explanation of what each "wp_" abbreviation represents (e.g., wp_ddd, wp_dad, wp_add). Discuss how these features are used by LightGBM and why they are effective in capturing shock wave characteristics.
- Comparison with Other ML Models (Figure 10, Table 3, Figure 12) is needed such as:
- In Figure 10, "LightGBM_updated" and "LightGBM_initial" are shown, but in Table 3 and Figure 12, only "LightGBM" is present. Clarify if "LightGBM_updated" corresponds to the LightGBM model being primarily discussed.
- The statement "with statistical significance (p < 0.05)" (line 403) needs to be supported with actual p-values or confidence intervals from statistical tests.
- Elaborate on why LightGBM outperforms the other models in this specific application. Discuss the advantages of its internal mechanisms (e.g., GOSS, EFB) in relation to the characteristics of the impact data.
- Refer to Figure 11 and 13:
- These figures appear to show similar data. Clarify if they represent the same results or different aspects. If they are redundant, consider combining them or removing one.
- Quantitative interpretation is needed; beyond showing the scatter, provide a quantitative analysis of the errors, perhaps with heatmaps or error distributions to show where the model performs better or worse.
- The highlights mention "real-time performance." Provide specific metrics or benchmarks for the inference time of the LightGBM model to substantiate this claim. How does it compare to the sampling frequency of the FBG sensors?
- Discuss limitations and future work; while a future work direction on irregular structures is mentioned, expand on the current limitations of the proposed method (e.g., impact energy range, material types, computational demands for training the model for new structures) and how these could be addressed.
There are several instances of awkward phrasing and grammatical errors throughout the paper. A thorough proofread by a native English speaker is highly recommended.
Round 2
Reviewer 1 Report
Comments and Suggestions for Authors
The manuscript has been revised in accordance with the reviewer's suggestions. I recommend its acceptance for publication.
Reviewer 2 Report
Comments and Suggestions for Authors
Accept.
Reviewer 3 Report
Comments and Suggestions for Authors
The authors are encouraged to revise Fig. 1 by enlarging the font size. With this minor revision, I recommend the manuscript for acceptance in MDPI Sensors.
